# Apremilast Improves Endothelial Glycocalyx Integrity, Vascular and Left Ventricular Myocardial Function in Psoriasis

**DOI:** 10.3390/ph15020172

**Published:** 2022-01-30

**Authors:** Ignatios Ikonomidis, George Pavlidis, Nikolaos Kadoglou, George Makavos, Konstantinos Katogiannis, Aikaterini Kountouri, John Thymis, Gavriella Kostelli, Irini Kapniari, Konstantinos Theodoropoulos, John Parissis, Pelagia Katsimbri, Evangelia Papadavid, Vaia Lambadiari

**Affiliations:** 12nd Department of Cardiology, Attikon University Hospital, Medical School, National and Kapodistrian University of Athens, 12462 Athens, Greece; geo_pavlidis@yahoo.gr (G.P.); gmakavos@hotmail.com (G.M.); kenndj89@gmail.com (K.K.); johnythg@gmail.com (J.T.); kosteligavriela@hotmail.com (G.K.); jparissis@yahoo.com (J.P.); 2Medical School, University of Cyprus, 2029 Nicosia, Cyprus; nikoskad@yahoo.com; 32nd Department of Internal Medicine, Research Unit and Diabetes Center, Attikon University Hospital, Medical School, National and Kapodistrian University of Athens, 12462 Athens, Greece; katerinak90@hotmail.com (A.K.); vlambad@otenet.gr (V.L.); 42nd Department of Dermatology and Venereology, Attikon University Hospital, Medical School, National and Kapodistrian University of Athens, 12462 Athens, Greece; kapniari_27@yahoo.gr (I.K.); theod28@gmail.com (K.T.); papadavev@yahoo.gr (E.P.); 54th Department of Internal Medicine, Attikon University Hospital, Medical School, National and Kapodistrian University of Athens, 12462 Athens, Greece; pelkats@gmail.com

**Keywords:** psoriasis, phosphodiesterase 4, apremilast, endothelial glycocalyx, perfused boundary region, myocardial deformation

## Abstract

The phosphodiesterase 4 inhibitor apremilast is used for the treatment of psoriasis. We investigated the effects of apremilast on endothelial glycocalyx, vascular and left ventricular (LV) myocardial function in psoriasis. One hundred and fifty psoriatic patients were randomized to apremilast (*n* = 50), anti-tumor necrosis factor-α (etanercept; *n* = 50), or cyclosporine (*n* = 50). At baseline and 4 months post-treatment, we measured: (1) Perfused boundary region (PBR), a marker of glycocalyx integrity, in sublingual microvessels with diameter 5–25 μm using a Sidestream Dark Field camera (GlycoCheck). Increased PBR indicates damaged glycocalyx. Functional microvascular density, an index of microvascular perfusion, was also measured. (2) Pulse wave velocity (PWV-Complior) and (3) LV global longitudinal strain (GLS) using speckle-tracking echocardiography. Compared with baseline, PBR_5–25 μm_ decreased only after apremilast (−12% at 4 months, *p* < 0.05) whereas no significant changes in PBR_5–25 μm_ were observed after etanercept or cyclosporine treatment. Compared with etanercept and cyclosporine, apremilast resulted in a greater increase of functional microvascular density (+14% versus +1% versus −1%) and in a higher reduction of PWV. Apremilast showed a greater increase of GLS (+13.5% versus +7% versus +2%) than etanercept and cyclosporine (*p* < 0.05). In conclusion, apremilast restores glycocalyx integrity and confers a greater improvement of vascular and myocardial function compared with etanercept or cyclosporine after 4 months.

## 1. Introduction

Psoriasis is a chronic immune-mediated skin disease associated with increased prevalence of atherosclerosis, elevated coronary microcirculation dysfunction and impaired vascular and left ventricular (LV) myocardial function [1]. Increased oxidative stress and inflammatory mediators including tumor necrosis factor (TNF)-α, interferon gamma (IFN-γ), interleukin (IL)-1β, IL-6, IL-12, IL-17, IL-22 and IL-23, released by activated T-cells subsets, monocytes and dendritic cells, are involved in the pathophysiology of psoriasis [2,3].

Endothelial glycocalyx is a complex mesh of sulfated proteoglycans, glycoproteins and associated glycosaminoglycans that covers the luminal side of vascular endothelial cells. It plays a pivotal role in vascular permeability by preventing the adhesion of leucocytes and platelets to the endothelium surface [4]. Pathophysiological conditions such as inflammation and increased oxidative stress are related to glycocalyx degradation resulting in endothelial dysfunction [5]. Damage of the glycocalyx integrity was proposed to promote early atherogenic processes [6,7,8] and is an independent and additive predictor to traditional risk factors for adverse outcome in subjects without overt cardiovascular disease [9].

Apremilast, an oral selective inhibitor of PDE4, is indicated for the treatment of patients with moderate-to-severe plaque psoriasis [10,11]. It is a safe and efficacious treatment for psoriatic patients, which combines skin and quality of life improvements [12]. Apremilast acts intracellularly by inhibiting cAMP degradation, leading to activation of protein kinase A (PKA). This enzyme induces cAMP responsive element binding protein (CREB) phosphorylation which can subsequently induces release of the anti-inflammatory IL-10. Additionally, PKA inhibits the nuclear factor kappa B (NF-κB) pathway, resulting in the suppression of major pro-inflammatory cytokines such as TNF-α, IL-2, IL-8, IL-23 and IFN-γ [13,14]. To date, there are initial evidence that apremilast is safe and beneficial immunomodulatory therapy in patients with coronavirus disease 2019 (COVID-19) pulmonary infection [15,16,17,18,19]. Anti-TNF-α biological agents and cyclosporine are also effective treatment regimens to improve psoriatic skin lesions through a strong inhibition of the inflammatory process [20]. On the other hand, other traditional drugs, such as methotrexate, have neutral effect on the cardiovascular risk in psoriasis, whereas biologic agents, such as IL-17A inhibitors, result in improvement of arterial and myocardial function [1]. However, any differential effects between apremilast, anti-TNF-α agents and cyclosporine on cardiovascular function have not been fully defined.

Because PDE4 determines several inflammatory pathways in psoriasis [21], we hypothesized that inhibition of PDE4 by apremilast would result in a greater improvement of endothelial glycocalyx integrity, vascular and LV myocardial function compared with TNF-α inhibitor or cyclosporine treatment. Thus, in the present randomized trial, we compared the effect of a 4-month treatment with PDE4 inhibitor (apremilast), anti-TNF-α agent (etanercept) or cyclosporine on endothelial glycocalyx, arterial stiffness and LV myocardial function. 

## 2. Results

Clinical characteristics including age, sex, traditional atherosclerotic risk factors and medication were similar among the three treatment groups (*p*
*>* 0.05; Table 1). In addition, at baseline all patients had similar PASI, body mass index (BMI), and similar indices of endothelial glycocalyx, vascular and LV myocardial function (*p*
*>* 0.05; Table 2). From all study patients, 44% (*n* = 66) achieved a reduction of PASI ≥ 75% and 28% (*n* = 43) a reduction of PASI ≥ 90% at 4 months. The percentage reduction of PASI score was similar in the three study arms (*p* = 0.435; Table 2). No changes were observed in BMI in the three treatment arms post-treatment (F = 0.525, *p* = 0.597; Table 2)

### 2.1. Association of PASI with Endothelial Glycocalyx, Vascular and LV Myocardial Function

At baseline, increased PASI was correlated with raised values of PBR_5–25 μm_ and reduced functional microvascular function (r = 0.37, *p* = 0.039 and r = −0.30, *p* = 0.044, respectively). Moreover, elevated PASI was associated with increased PWV, impaired (less negative) GLS and decreased %dpTw-Utw_MVO_ (r = 0.34, *p* = 0.045, r = 0.38, *p* = 0.023, and r = −0.42, *p* = 0.014, respectively).

### 2.2. Interrelation of Endothelial Glycocalyx, Vascular and LV Myocardial Function

In the whole study population, impaired endothelial glycocalyx barrier function, as estimated by increased PBR_5–25 μm_, was inversely associated with impaired microvascular perfusion, as assessed by reduced functional microvascular density (r = −0.48, *p* < 0.001). Furthermore, PBR_5–25 μm_ was directly correlated with PWV, GLS and PWV/GLS (r = 0.31, *p* = 0.042, r = 0.40, *p* = 0.001, and r = 0.28, *p* = 0.043, respectively) and inversely related to %dpTw-Utw_MVO_ (r = −0.27, *p* = 0.046).

### 2.3. Effects of Treatment on Endothelial Glycocalyx, Vascular and LV Myocardial Deformation

#### 2.3.1. Endothelial Glycocalyx and Microvascular Perfusion

In the overall study population, there were no changes in PBR_5–25 μm_ (*p* = 0.264) and in functional microvascular density (*p* = 0.180) post-treatment after adjusted for age, sex, PASI, risk factors, and medication (Table 2). However, there was a significant interaction between the type of treatment and changes in PBR (F = 9.7, *p* for interaction <0.001) and functional microvascular density (F = 9.1, *p* for interaction <0.001) post-treatment.

Compared to baseline, patients on apremilast treatment had reduced PBR_5–25 μm_ (*p* < 0.001; Figure 1a) indicating a restored glycocalyx and increased functional microvascular density (*p* = 0.023; Figure 1b) at 4 months. In the etanercept and cyclosporine groups, there were no significant changes in PBR_5–25 μm_ (*p* = 0.917 and *p* = 0.293) and functional microvascular density (*p* = 0.783 and *p* = 0.525) between baseline and post-treatment. 

Compared with etanercept and cyclosporine, apremilast resulted in a greater decrease of PBR_5–25 μm_ (−12% versus +0.1% versus +2%, respectively; *p* < 0.001) and in a greater increase of functional microvascular density (+14% versus +1% versus −1%, respectively; *p* < 0.001) at 4-month treatment.

#### 2.3.2. Vascular Function

In all patients, there were no significant differences in arterial stiffness as assessed by PWV (*p* = 0.802) and in central systolic blood pressure (cSBP; *p* = 0.444) post-treatment after adjusted for age, sex, PASI, risk factors, and medication (Table 2). However, there was a significant interaction between the type of treatment and changes in PWV and cSBP post-treatment (F = 8.9, *p* for interaction <0.001 and F = 8.8, *p* for interaction <0.001; Table 2).

Compared to baseline, patients treated with apremilast had reduced PWV (*p* = 0.044) and decreased cSBP (*p* = 0.023). In the etanercept group, there were no differences in PWV (*p* = 0.356) and cSBP (*p* = 0.241). Post cyclosporine treatment, patients showed significant increased values of PWV (*p* = 0.045) and cSBP (*p* = 0.031) compared with baseline. 

Compared with etanercept and cyclosporine, apremilast resulted in a higher reduction of PWV (−9% versus −3% versus +9%, respectively; *p* < 0.001) and cSBP (−8% versus −2% versus +8%, respectively; *p* < 0.001) at 4 months.

#### 2.3.3. Myocardial Function 

All patients had improved LV myocardial function, namely GLS (*p* = 0.008), peak twisting (*p* = 0.03), untwisting at mitral valve opening (*p* = 0.042) and percent difference between peak LV twisting and untwisting at MVO (*p* = 0.036) post-treatment after adjusted for age, sex, PASI, risk factors, and medication (Table 2). However, there was a significant interaction between the type of treatment and changes in GLS (F = 6.8, *p* for interaction = 0.012), peak twisting (F = 5.4, *p* for interaction = 0.025), untwisting at mitral valve opening (F = 4.5, *p* for interaction = 0.04) and percent difference between peak LV twisting and untwisting at MVO (F = 5.7, *p* for interaction = 0.022) post-treatment (*p* < 0.05; Table 2). 

Psoriatic patients treated with apremilast or etanercept had improved GLS (*p* = 0.007 and *p* = 0.02), peak twisting (*p* = 0.021 and *p* = 0.026), untwisting at mitral valve opening (*p* = 0.037 and *p* = 0.048), and percent difference between peak LV twisting and untwisting at MVO (*p* = 0.029 and *p* = 0.035) after 4 months compared with baseline (Table 2). In contrast, patients treated with cyclosporine had no significant changes in GLS (*p* = 0.568), peak twisting (*p* = 0.337), untwisting at mitral valve opening (*p* = 0.923) and percent difference between peak LV twisting and untwisting at MVO (*p* = 0.415) post-treatment compared to baseline. 

Compared with etanercept and cyclosporine, apremilast showed a greater increase of GLS (+13.5% versus +7% versus +2%, respectively; *p* = 0.017). Furthermore, apremilast resulting in a greater improvement of PWV/GLS (−19% versus −8% versus +5%, respectively; *p* = 0.019), peak twisting (+18% versus +14% versus +3%, respectively; *p* = 0.028), untwisting at mitral valve opening (+12% versus +8% versus +1%, respectively; *p* = 0.044) and %dpTw-Utw_MVO_ (+18% versus +12% versus +3%, respectively; *p* = 0.027) than etanercept and cyclosporine.

### 2.4. Association of the ΔPASI ≥ 75% with Markers of Vascular and Myocardial Function

In the whole study population, patients with ΔPASI ≥ 75% (*n* = 66 out of 150 patients) had similar baseline PWV and GLS values to those with ΔPASI < 75% (10 ± 2.9 m/s versus 10.4 ± 2.3 m/s, *p* = 0.692 and −16.9 ± 3.6% versus −17.3 ± 4.2%, *p* = 0.803, respectively). In contrast, 4 months post-treatment, patients with ΔPASI ≥ 75%, showed a greater improvement of PWV and GLS compared to those with ΔPASI < 75%, (9.5 ± 2.4 m/s versus 10.3 ± 2.1 m/s, *p* = 0.044 and −19 ± 4.5% versus −18.1 ± 3.9%, *p* = 0.037, respectively). Nonetheless, there was no a significant interaction between the type of treatment and ΔPASI ≥ 75%, regarding changes in PWV (*p* for interaction = 0.495) or GLS (*p* for interaction = 0.525) post-treatment. No significant differences between responders and non-responders to treatment were observed for the remaining examined markers.

### 2.5. Association between Changes of Endothelial Glycocalyx Markers with Changes in Vascular and LV Myocardial Function Post-Treatment

Changes of PBR_5–25 μm_ and functional microvascular density post-apremilast treatment correlated with a concomitant reduction of PWV (r = 0.47, *p* < 0.001 and r = −0.43, *p* < 0.001, respectively) and cSBP (r = 0.44, *p* < 0.001 and r = −0.39, *p* = 0.002, respectively) and with an improvement of GLS (r = −0.31, *p* = 0.019 and r = 0.29, *p* = 0.042, respectively).

## 3. Discussion

In the present study of patients with psoriasis, 4-month treatment with the inhibitor of PDE4, apremilast, improved endothelial glycocalyx barrier function, as assessed by PBR_5–25 μm,_ and microvascular perfusion, as estimated by functional microvascular density. Furthermore, apremilast achieved a greater reduction of arterial stiffness, as assessed by PWV, and a greater improvement in LV longitudinal myocardial deformation and twisting-untwisting compared with the anti-TNF-α agent, etanercept, and cyclosporine. The improvement of PBR_5–25 μm_ and functional microvascular density post-apremilast treatment associated with the concomitant improvement of arterial stiffness and LV myocardial deformation in our study. 

### 3.1. Effects of Treatment on Endothelial Glycocalyx and Microvascular Perfusion

In a recent study, we have shown that PBR, an accurate index of endothelial glycocalyx dimensions, together with functional microvascular density, are impaired in patients with psoriasis compared to healthy controls [22]. This finding is suggestive of the presence of endothelial glycocalyx damage and perfusion defects in patients with chronic inflammatory diseases such as psoriasis. Indeed, inflammatory-mediated glycocalyx degradation by pro-inflammatory cytokines such as TNF-α, IL-1, IL-6, and IL-8 leads to alterations in endothelial permeability with consequent transcapillary water and protein hyperpermeability, interstitial fluid shift and generalized edema [23,24]. Hence, glycocalyx injury can be responsible for a number of several clinical conditions characterized by the excess cytokines release, such as severe acute respiratory syndrome coronavirus 2 (SARS-CoV-2). Actually, higher PBR values were observed in patients with COVID-19 on mechanical ventilation compared to non-ventilated and controls [25]. Interestingly, recent study showed that apremilast has distinct anti-inflammatory effects through the suppression of TNF-α-induced release of significant endothelial pro-inflammatory factors such as granulocyte-macrophage colony-stimulating factor, vascular cell adhesion molecule-1 and matrix metalloprotein-9 in human umbilical vein endothelial cells. The suppression of pro-inflammatory molecules occurs partially via a signaling pathway dependent on PDE4 inhibition [26]. Furthermore, apremilast improves oxidized low-density lipoprotein-induced endothelial dysfunction via the rescue of Krüppel like factor-6 expression, suggesting a potential role for apremilast in the treatment of atherosclerotic cardiovascular disease (Figure 2) [27]. Due to the unique mechanism of action targeting the early stage of inflammatory response, inducing upstream inhibition of multiple molecular signaling pathways, including IL-6 production, apremilast may be an effective therapeutic option for the early phases of SARS-CoV-2 pneumonia [28]. Previous study has shown that inhibition of IL-6, a signature cytokine of COVID-19, is associated with improved glycocalyx integrity and thus reduced vascular permeability in the setting of cytokine overexpression [29]. Thus, the effects of apremilast on endothelial glycocalyx may also explain its reported beneficial action in COVID-19 [28].

On the other hand, apremilast presents beneficial metabolic effects by reducing body weight, increasing lipolysis, reducing lipid deposition in liver, and improving insulin sensitivity, especially in patients with increased glycated hemoglobin A1c with or without concomitant antidiabetic medication (Figure 2) [30,31]. We have previously demonstrated that insulin resistance, as assessed by insulin sensitivity index and Matsuda index, was associated with PBR suggesting its detrimental effect on glycocalyx integrity [32]. Indeed, in the present study we demonstrated that endothelial glycocalyx and microvascular perfusion were improved only after apremilast treatment whereas no significant changes in of these markers were observed after etanercept or cyclosporine in patients with psoriasis. This finding supports that treatment with apremilast enhances glycocalyx barrier integrity not only by its anti-inflammatory action but also by its metabolic effect on endothelial glycocalyx. However, a recent study showed that the improvement of disease activity in psoriatic arthritis is independent of the abdominal fat reduction, observed post-apremilast treatment, suggesting that apremilast may act mainly through direct immunological mechanisms and to a less extent through metabolic effects [33]. 

### 3.2. Effects of Treatment in Vascular and Myocardial Function 

As we have shown in a recent study, endothelial glycocalyx impairment was associated with increased arterial stiffness, as assessed by PWV, impaired LV myocardial deformation, as estimated by decreased global longitudinal strain and LV twisting-untwisting, and impaired ventricular-arterial interaction, as estimated by PWV/GLS ratio [22]. Elevated inflammatory burden and oxidative stress resulting in vascular and LV myocardial dysfunction may explain the link between increased arterial stiffness and impaired myocardial performance in patients with psoriasis [1]. In the current study, there was a significant reduction of PWV and central SBP after apremilast treatment and not after etanercept. Furthermore, we observed a detrimental effect of cyclosporine on arterial stiffness in accordance with the reported adverse effects of cyclosporine on vascular function [34]. Additionally, psoriatic patients showed a greater improvement in GLS, PWV/GLS ratio, and LV twisting-untwisting after apremilast than etanercept or cyclosporine despite a similar reduction of PASI. In our study, the improvement in vascular and myocardial function was related with a respective improvement in endothelial glycocalyx integrity. These findings may be interpreted by the downregulated expression of multiple pro-inflammatory mediators such as TNF-α, IL-6, IL-8, IL-17 and IL-23 post-apremilast treatment, which are linked to increased arterial stiffness, oxidative stress, LV myocardial dysfunction and cardiac remodeling [1]. Previous studies have shown that inhibition of IL-6, IL-17 and IL-23 are related with improved vascular function and cardiac deformation [1,29]. In contrast, treatment with TNF-α inhibitors causes inhibition of specific inflammatory pathways determined only by TNF-α activity resulting to narrow anti-inflammatory action compared to the wider range of anti-inflammatory action by apremilast. Interestingly, previous studies have demonstrated that apremilast can prevent carfilzomob- and doxorubicin-induced cardiac inflammation in rats by inhibition of oxidative stress-mediated activation of NF-κB and mitogen activated protein kinase signaling pathways [35,36]. Furthermore, inhibition of PDE4 attenuates artery injury by the inhibition of platelet-neutrophil interactions and may improve atherosclerotic lesions [37,38]. The above studies suggest a wider effect of apremilast treatment on oxidative and inflammatory pathways compared to anti-TNF-α treatment. 

### 3.3. Limitations 

There are some limitations to our study. Firstly, the study design does not allow one to investigate the causality for the changes in endothelial glycocalyx integrity, microvascular perfusion and LV myocardial deformation post-apremilast treatment. Additionally, the Sideview Dark Field camera permit a non-invasive estimation of endothelial glycocalyx based on the red blood cells column distribution. Thus, the limitation of this method is the indirect assessment of endothelial glycocalyx dimensions. Finally, our study was a single-center trial and not blinded to patients. Further prospective large-scale clinical trials are required to determine whether improvement of endothelial glycocalyx barrier function, arterial stiffness, and LV myocardial function is maintained in the long run. Moreover, the potential effect of apremilast on reducing cardiovascular events requires longer prospective studies and comparison with other drugs such as methotrexate, other TNF inhibitors, IL-17 and/or IL-23 antagonists.

## 4. Materials and Methods

### 4.1. Study Protocol

One hundred and eighty two psoriatic patients, who were referred to the outpatient psoriasis clinic, was assessed for eligibility by the attending dermatologists (I.K., K.T., and E.P.). The recruitment took place between April of 2020 and March of 2021. Inclusion criteria were male or female patients aged 18 to 75 years with plaque-type psoriasis. Exclusion criteria were pregnancy or breastfeeding, psoriatic arthritis, inflammatory bowel disease, history of coronary artery disease (CAD), moderate or severe valve disease, primary cardiomyopathies, congestive heart failure, chronic kidney disease (CKD; estimated glomerular filtration rate [eGFR] ≤60 mL/min/1.73 m^2^), severe liver insufficiency, latent or active tuberculosis, underweight subjects (BMI < 18.5 Kg/m^2^), and malignancies. Of the 182 enrolled patients, 32 patients were excluded from the study either due to preexisting or suspected CAD (*n* = 5) and heart failure (*n* = 4), or because of inadequate echocardiography images for analysis (*n* = 15) and 8 patients declined to participate (Appendix A). Hence, 150 patients were randomized to receive apremilast 30 mg twice daily, after an initial 5-day titration period (*n* = 50), anti-tumor necrosis factor-α, namely etanercept 50 mg subcutaneous, two days per week (*n* = 50), or cyclosporine 2.5–3 mg/Kg daily (*n* = 50) for 4 months. According to the standard of care of patients with moderate to severe plaque psoriasis, biologic treatments, such as anti-TNF-α agents, and oral medications, including cyclosporine, would be used to start therapy, as previously published [1,20]. Randomization was performed by an attending dermatologist (E.P.) using a random number table as reproduced from the online randomization software http://www.graphpad.com/quickcalcs/index.cfm (accessed on 6 April 2020). In all patients, we assessed endothelial, vascular and LV myocardial function on the same day at baseline and 4 months after the initiation of treatment. Patients and clinicians prescribing the treatment were blinded to the results of endothelial, vascular and cardiac function examinations. There were no losses and exclusions after randomization in the three study groups. The severity of psoriasis was estimated using the Psoriasis Area and Severity Index (PASI) [39]. Hence, PASI was calculated at baseline and post-treatment to estimate the effectiveness of treatment. We also report the number of patients who showed a reduction of ≥75% and ≥90% in PASI score (ΔPASI ≥ 75 and ΔPASI ≥ 90) post-treatment. The study protocol was approved by the Institute’s Ethics Committee and was conducted according to the ethical standards of the World Medical Association Declaration of Helsinki. Written informed consent was obtained from all patients prior to participation. 

### 4.2. Primary and Secondary Endpoints

The primary outcome was changes in endothelial glycocalyx integrity and functional microvascular density after 4 months treatment with apremilast compared with etanercept and cyclosporine treatment. Secondary outcomes were changes in pulse wave velosity, global LV longitudinal strain and LV twisting and untwisting post-apremilast treatment compared with etanercept and cyclosporine treatment. 

### 4.3. Imaging of Microcirculation

In vivo evaluation of the perfused boundary region (PBR, μm) of the sublingual microvasculature was performed with a Sidestream Dark Field (SDF) camera (Microscan, GlycoCheck, Microvascular Health Solutions Inc., Salt Lake City, UT, USA). The SDF camera uses green light-emitting diode (540 nm) to detect the hemoglobin of circulating red blood cells (RBCs) within the microvessels [9]. It is inserted under the tongue and captures >3000 microvessels segments with a lumen diameter ranging from 5 to 25 μm. The images are recorded and analyzed automatically by the GlycoCheck software. The median RBC column width and the total perfused diameter of microvessels is assessed by the software. The PBR is calculated using the following formula: (perfused diameter—median RBC column width)/2. Increased PBR values indicate a deeper penetration of RBCs into the endothelial surface and reflect a damaged glycocalyx [40]. Degradation of glycocalyx has been linked to inflammatory conditions and atherosclerosis [5]. The exam lasts about 3 min, has satisfactory reproducibility and is recommended as a valid method for the evaluation of endothelial function [4]. The inter- and intra-observer variability of PBR was 5.2% and 4.3%, respectively. 

Microvascular density was estimated by the GlycoCalyx software that identifies by contrast between RBC and the background and automatically records at least 3000 microvessel segments. After the acquisition, line markers are placed every 10 μm along the length of microvessels. On the first frame of each video recording session, a total of 21 line markers are placed every 0.5 μm of the microvessel segments. Only those microvessels that have contrast enhancement >60% of the 21 line markers are considered as functional microvessel segments. Functional microvascular density is estimated by the equation: (number of functional microvessel segments × 10)/tissue area recorded, and is considered as an index of microvascular perfusion [22,40]. 

### 4.4. Arterial Stiffness

Carotid to femoral pulse wave velocity (PWV, m/s), which is an index of arterial stiffness and underlying arteriosclerosis, was measured using a previously published methodology (Complior SP, ALAM Medical, Vincennes, France) [22]. PWV was calculated by dividing the distance from the carotid to the femoral point by the transit time. 

### 4.5. Echocardiography

Echocardiography studies were performed using a Vivid Ε95 (GE Medical Systems, Horten, Norway) ultrasound system. All studies were digitally stored in a computerized station (EchoPac GE 203) and were analyzed by two investigators (I.I. and G.P.), blinded for clinical and laboratory data.

### 4.6. Two-Dimensional Strain Measurements

In all patients, two-dimensional strain was measured by speckle tracking analysis from the apical four, two and three chamber views (frame rate: 70–80/s) utilizing a dedicated software (EchoPac PC 203, GE Healthcare, Chicago, IL, USA). We calculated the LV global longitudinal strain (GLS, %) as previously published [1]. The inter- and intra-observer variability of GLS was 7% and 10%, respectively.

In addition, we calculated the ratio of pulse wave velocity to LV global longitudinal strain (PWV/GLS, m/s%) as an index of ventricular-arterial interaction, as previously published [1].

Left ventricular peak twisting (pTw, °) and untwisting at mitral valve opening (Utw_MVO_, °), were estimated from the parasternal short axis views at the basal and the apical level in speckle-tracking mode. The inter- and intra-observer variability for LV peak twisting and untwisting was ≤8% and ≤10%, respectively [22]. Moreover, the degree of LV untwisting during diastole was measured as the percentage difference between peak twisting and untwisting at mitral valve opening (%dpTw-Utw_MVO_), at peak (%dpTw-Utw_PEF_), and at the end of mitral E wave (%dpTw-Utw_EDF_) according to our previously published methodology [22].

### 4.7. Statistical Analysis

We planned to investigate the percentage change (Δ) of PBR_5–25 μm_ post treatment from independent control (patients treated with cyclosporine) and experimental subjects (patients treated with apremilast) with 1 control per 1 experimental subject. In a pilot study of 10 patients treated with cyclosporine and 10 treated with apremilast, the response within each subject group was normally distributed with standard deviation 0.25. The true difference between patients on cyclosporine and those on apremilast in the mean values of PBR_5–25 μm_ was 14%. Hence, we would need to study 50 patients with psoriasis treated with cyclosporine and 50 treated with apremilast to be able to reject the null hypothesis that the study population means for ΔPBR_5–25 μm_ post treatment of the cyclosporine and apremilast groups are equal with probability (power) 0.8 and type I error probability is 0.05. 

Statistical analyses were done with Statistical Package for Social Sciences version 22.0 for Windows (SPSS Statistics, Inc., Chicago, IL, USA). Scale variables are expressed as mean ± standard deviation (SD). Categorical variables are presented as percentages of the study population and were compared by the chi-square test. Continuous variables were tested by the Kolmogorov-Smirnov test to assess the normal distribution. In case of non-normal distribution, variables were analyzed after transformation into ranks.

We applied intention-to-treat analysis. Analysis of variance (ANOVA) for repeated measurements was performed for measurements of the examined markers at baseline and 4 months after treatment used as a within-subject factor, and for the effect of treatment (apremilast, etanercept, and cyclosporine) as a between-subject factor. The F and *p* values of the interaction between time of measurement of the examined variables and type of treatment were calculated. The F and *p* values of the comparison between treatments were also calculated. When the sphericity assumption, as estimated by Mauchly’s test, was not met, the Greenhouse-Geisser correction was used. Post hoc multiple comparisons were applied with Bonferroni correction. Baseline variables that were of clinical significance, namely age, sex, PASI, risk factors, and medication, were included in multivariate models as covariates. The percentage changes of the examined markers post-treatment between the three study groups were also analyzed using ANOVA. Simple correlations between continuous variables were determined using parametric (Pearson) or non-parametric (Spearman) correlation coefficients. All statistical tests were two-tailed and values of *p* < 0.05 were considered significant. 

## 5. Conclusions 

In psoriasis, apremilast confers a greater improvement of endothelial glycocalyx integrity, microvascular perfusion, arterial elasticity and LV myocardial function compared with etanercept or cyclosporine treatment, suggesting a favorable profile of PDE4 inhibition on cardiovascular function. These findings suggest that apremilast is safe in patients with increased cardiovascular risk.

## Figures and Tables

**Figure 1 pharmaceuticals-15-00172-f001:**
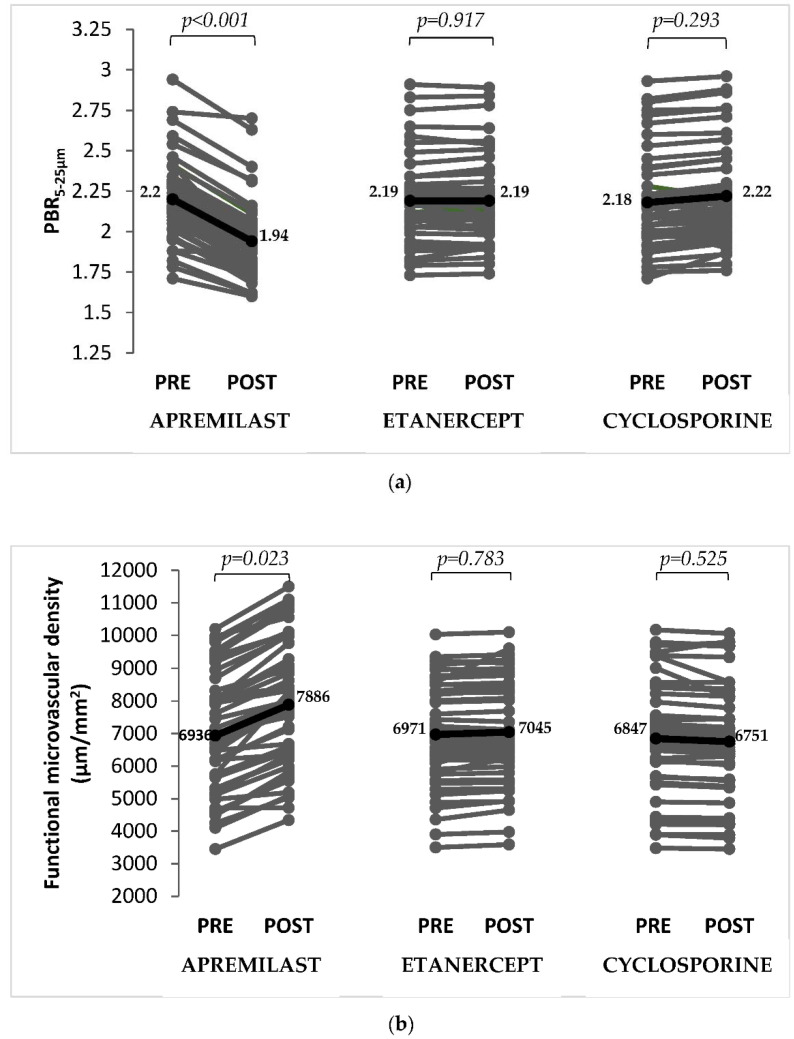
Perfused boundary region in sublingual microvessels with diameter 5–25 μm (PBR_5–25 μm_) and functional microvascular density in the three study groups. (**a**) PBR_5–25 μm_ decreased in patients treated with apremilast, whereas there was no significant changes in patients treated with etanercept or cyclosporine. (**b**) Functional microvascular density increased in patients treated with apremilast, whereas no significant changes were observed in patients treated with etanercept or cyclosporine. Solid black dots indicate mean values.

**Figure 2 pharmaceuticals-15-00172-f002:**
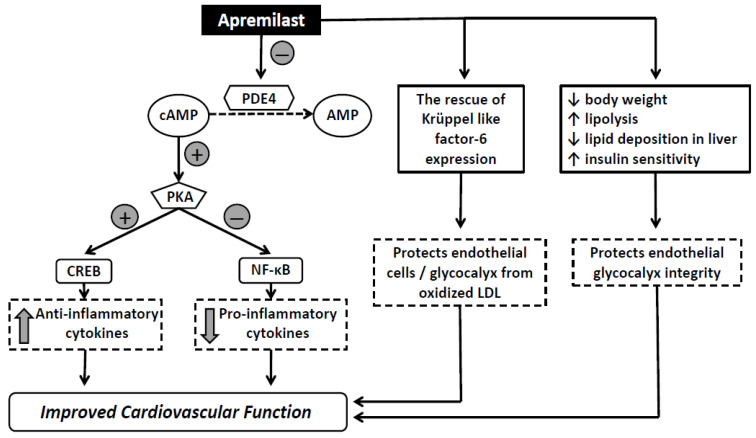
Schematic representation of the molecular mechanism of action of apremilast and the possible effects on cardiovascular system. The inhibition of phosphodiesterase 4 (PDE4) by apremilast increases intracellular levels of cAMP leading to activation of protein kinase A (PKA). This enzyme induces cAMP responsive element binding protein (CREB) phosphorylation which subsequently induces release of anti-inflammatory cytokines. Moreover, PKA inhibits the nuclear factor kappa B (NF-κB) pathway, resulting in the suppression of major pro-inflammatory cytokines. Apremilast improves oxidized low-density lipoprotein (LDL)-induced endothelial dysfunction via the rescue of Krüppel like factor-6 expression and presents beneficial metabolic effects, suggesting a potential role for apremilast in the improvement of cardiovascular function.

**Table 1 pharmaceuticals-15-00172-t001:** Clinical characteristics of the study population.

	All Patients (*n* = 150)	Apremilast (*n* = 50)	Etanercept (*n* = 50)	Cyclosporine (*n* = 50)	*p*-Value
Age, years	51 ± 12	51 ± 11	51 ± 13	50 ± 10	0.820
Sex (male/female), *n* (%)	90/60 (60/40)	30/20 (60/40)	29/21 (58/42)	31/19 (62/38)	0.684
BMI, kg/m^2^	30 ± 5	30 ± 4	31 ± 6	30 ± 5	0.992
Duration of disease, years	16 ± 11	17 ± 12	16 ± 11	14 ± 9	0.528
PASI	12 ± 2.3	12 ± 2.4	13 ± 2.6	12 ± 2.1	0.922
Risk factors, *n* (%)					
Hypertension	54 (36)	19 (38)	18 (36)	17 (34)	0.918
Dyslipidemia	53 (35)	18 (36)	19 (38)	16 (32)	0.467
Diabetes Mellitus	23 (15)	8 (16)	7 (14)	8 (16)	0.843
Current smoking	80 (53)	26 (52)	26 (52)	28 (56)	0.598
Family history CAD	21 (14)	8 (16)	6 (12)	7 (14)	0.693
Μedication, *n* (%)					
Beta blockers	28 (19)	10 (20)	9 (18)	9 (18)	0.923
CCBs	45 (30)	16 (32)	16 (32)	13 (26)	0.634
ACEI/ARBs	48 (32)	17 (34)	16 (32)	15 (30)	0.847
Diuretics	30 (20)	10 (20)	11 (22)	9 (18)	0.899
Statins	53 (35)	18 (36)	19 (38)	16 (32)	0.467
Fibrate	5 (3)	2 (4)	2 (4)	1 92)	0.988
Antidiabetic agents	23 (15)	8 (16)	7 (14)	8 (16)	0.843

Data are expressed as number (%) and mean values ± standard deviation. Continuous variables were compared with the paired Student t test. Binary variables were compared with the chi-square test. BMI, body mass index; PASI, psoriasis area and severity index; CAD, coronary artery disease; CCB, calcium channel blocker; ACEI, angiotensin-converting enzyme inhibitor; ARB, angiotensin receptor blocker.

**Table 2 pharmaceuticals-15-00172-t002:** Changes in indices of endothelial glycocalyx barrier function, microvascular perfusion, arterial stiffness and echocardiographic markers of LV myocardial function in the 3 treatment groups during the study period.

		All Patients (*n* = 150)	Apremilast (*n* = 50)	Etanercept (*n* = 50)	Cyclosporine (*n* = 50)
ΔPASI75, %		44	46	44	42
ΔPASI90, %		28	32	28	26
BMI, Kg/m^2^	Baseline	30 ± 5	30 ± 4	31 ± 6	30 ± 5
	4 months	30 ± 5	29 ± 4	31 ± 5	31 ± 5
	Δ%	−1	−3	−1	+3
PBR_5–25 μm_	Baseline	2.19 ± 0.25	2.20 ± 0.25	2.19 ± 0.25	2.18 ± 0.26
	4 months	2.12 ± 0.25	1.94 ± 0.24 ^‡^	2.19 ± 0.27	2.22 ± 0.25 ^†^
	Δ%	−3	−12	+0.1	+2
Functional microvascular density, μm/mm^2^	Baseline	6918 ± 2413	6936 ± 2448	6971 ± 2259	6847 ± 2532
	4 months	7227 ± 2034	7886 ± 2161 ^‡^	7045 ± 2114	6751 ± 1828 ^†^
	Δ%	+4	+14	+1	−1
PWV, m/s	Baseline	10.2 ± 2.4	10.4 ± 3	10.3 ± 2.1	9.9 ± 2
	4 months	10.1 ± 2.2	9.5 ± 2.4 ^‡^	10 ± 2.2	10.8 ± 1.9 ^†‡^
	Δ%	−1	−9	−3	+9
cSBP, mmHg	Baseline	133 ± 26	132 ± 31	135 ± 29	132 ± 19
	4 months	132 ± 22	122 ± 20 ^‡^	132 ± 22	143 ± 23 ^†‡^
	Δ%	−1	−8	−2	+8
GLS, %	Baseline	−17.2 ± 4	−17.1 ± 3	−17.3 ± 4	−17.2 ± 3
	4 months	−18.5 ± 4 ^§^	−19.4 ± 3 ^§^	−18.5 ± 3 ^‡^	−17.5 ± 4 *
	Δ%	+7	+13.5	+7	+2
PWV/GLS, m/s%	Baseline	−0.59 ± 0.20	−0.61 ± 0.21	−0.59 ± 0.20	0.58 ± 0.19
	4 months	−0.55 ± 0.19	−0.49 ± 0.17 ^‡^	−0.54 ± 0.19	0.61 ± 0.22 *
	Δ%	−7	−19	−8	+5
pTw, °	Baseline	14.6 ± 6	14.9 ± 6.1	14.3 ± 5.4	14.8 ± 6.3
	4 months	16.5 ± 6.2 ^‡^	18.2 ± 6.8 ^‡^	16.3 ± 5.6 ^‡^	15.2 ± 5.9 *
	Δ%	+13	+18	+14	+3
Utw_MVO_, °	Baseline	9.8 ± 4.6	9.8 ± 4.8	9.7 ± 4.3	9.8 ± 4.7
	4 months	10.5 ± 4.8 ^‡^	11 ± 5.2 ^‡^	10.5 ± 4.5 ^‡^	9.9 ± 4.6 *
	Δ%	+7	+12	+8	+1
dpTw-Utw_MVO_, %	Baseline	33 ± 10	34 ± 12	32 ± 10	34 ± 9
	4 months	37 ± 10 ^‡^	41 ± 8 ^‡^	36 ± 10 ^‡^	35 ± 11 *
	Δ%	+12	+18	+12	+3

Data are presented as mean values ± standard deviation. Δ% indicates percent changes from baseline. ΔPASI75, percentage reduction of ≥75% in psoriasis area and severity index (PASI) score; ΔPASI90, percentage reduction of ≥90% in PASI score; BMI, body mass index; PBR_5–25 μm_, perfused boundary region in sublingual microvessels with diameter 5–25 μm; PWV, pulse wave velocity; cSBP, central systolic blood pressure; GLS, global longitudinal strain; PWV/GLS, ventricular-arterial interaction (pulse wave velocity to global longitudinal strain ratio); pTw, peak twisting; Utw_MVO_, peak untwisting at the time of mitral valve opening; dpTw-Utw_MVO_, percent difference between peak twisting and untwisting at mitral valve opening. * *p* < 0.05, ^†^
*p* < 0.001 for time × treatment interaction obtained by repeated-measures ANOVA. ^‡^* p* < 0.05, ^§^* p* < 0.01 for comparisons of 4 months versus baseline by ANOVA using post hoc analysis with Bonferroni correction.

## Data Availability

The datasets generated and/or analyzed during the current study are not publicly available due to information that could compromise the privacy of research participants but are available from the corresponding author upon reasonable request. All of other data is contained within the article.

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
