# Peer review of "Apremilast Improves Endothelial Glycocalyx Integrity, Vascular and Left Ventricular Myocardial Function in Psoriasis"

_pharmaceuticals, 2022, doi:10.3390/ph15020172_

Round 1

Reviewer 1 Report

The Authors assessed the impact of apremilast treatment on endothelial glycocalyx, vascular and left ventricular and myocardial function in psoriasis. They compare the results with standard treatment (cyclosporine, etanercept).

The results are quite interesting, however there are some minor issues that need to be addressed before publication. 

  • The introduction lacks information on the impact of other traditional drugs, such as methotrexate or biological drugs, such as IL-17A inhibitors, on the cardiovascular risk in psoriasis. The choice of drugs compared with apremilast in the study should be justified in the discussion or methodology sections.
  • The efficacy of individual treatments should be presented as the percentage of patients who achieved a delta PASI75 and 90 response. 
  • Were there differences between responders and non-responders for individual drugs?
  • What were the doses of the drugs used?
  • "These findings suggest that apremilast may be useful for the prevention and treatment of cardiovascular disease." This conclusion is too general and not fully supported by the results. It can be concluded that apremilast is safe in patients with increased cardiovascular risk. The potential effect on reducing cardiovascular events requires longer prospective studies and comparison with other drugs such as methotrexate, other TNF inhibitors, IL-17 and / or IL-23 antagonists. This should be added to the limitations of the study.

Author Response

Response to Reviewer 1 Comments

We would like to thank this Reviewer for his/her constructive comments which helped us improve our manuscript.

Point 1: The introduction lacks information on the impact of other traditional drugs, such as methotrexate or biological drugs, such as IL-17A inhibitors, on the cardiovascular risk in psoriasis. The choice of drugs compared with apremilast in the study should be justified in the discussion or methodology sections.

Response 1: We apologize for the lack of information in the Introduction about the impact of other traditional drugs, such as methotrexate or biological drugs on the cardiovascular risk in psoriasis. Following the reviewer’s suggestion, we have now added the following phrase:

Page 2, lines 68-71: “On the other hand, other traditional drugs, such as methotrexate, have neutral effect on the cardiovascular risk in psoriasis, whereas biologic agents, such as IL-17A inhibitors, result in improvement of arterial and myocardial function [1].”

Also, we have justified the choice of drugs in the Materials and Methods section, as follows:

Page 9, lines 335-7: “According to the standard of care of patients with moderate to severe plaque psoriasis, biologic treatments, such as anti-TNF-α agents, and oral medications, including cyclosporine, would be used to start therapy, as previously published [1,20].”

Point 2: The efficacy of individual treatments should be presented as the percentage of patients who achieved a delta PASI75 and 90 response.

Response 2: We acknowledge the reviewer’s comment. Following the reviewer’s suggestion, in the Table 2 of the revised version (page 3) we removed PASI values and we have now added ΔPASI75 and ΔPASI90.

Also, we have added in the Results section the following sentence:

Page 2, lines 84-7: “From all study patients, 44% (n=66) achieved a reduction of PASI ≥75% and 28% (n=43) a reduction of PASI ≥90% at 4 months. The percentage reduction of PASI score was similar in the 3 study arms (p=0.435; Table 2).”

Point 3: Were there differences between responders and non-responders for individual drugs?

Response 3: Following the reviewer’s request, we have now added in the Results section the following paragraph:

Page 6, lines 193-203: “2.4. Association of the ΔPASI≥75%, with markers of vascular and myocardial function

In the whole study population, patients with ΔPASI≥75%, (n=66 out of 150 patients) had similar baseline PWV and GLS values to those with ΔPASI<75%, (10±2.9m/s versus 10.4±2.3m/s, p=0.692 and -16.9±3.6% versus -17.3±4.2%, p=0.803, respectively). In contrast, 4 months post-treatment, patients with ΔPASI≥75%, showed a greater improvement of PWV and GLS compared to those with ΔPASI<75%, (9.5±2.4m/s versus 10.3±2.1m/s, p=0.044 and -19±4.5% versus -18.1±3.9%, p=0.037, respectively). Nonetheless, there was no a significant interaction between the type of treatment and ΔPASI≥75%, regarding changes in PWV (p for interaction=0.495) or GLS (p for interaction=0.525) post-treatment. No significant differences between responders and non-responders to treatment were observed for the remaining examined markers.”

Point 4: What were the doses of the drugs used?

Response 4: Following the reviewer’s request, we are now providing the doses of the drugs used, as follows:

Page 9, lines 331-4: “Hence, 150 patients were randomized to receive apremilast 30 mg twice daily, after an initial 5-day titration period (n=50), anti-tumor necrosis factor-α, namely etanercept 50 mg subcutaneous, two days per week (n=50), or cyclosporine 2.5-3 mg/Kg daily (n=50) for 4 months.”

Point 5: "These findings suggest that apremilast may be useful for the prevention and treatment of cardiovascular disease." This conclusion is too general and not fully supported by the results. It can be concluded that apremilast is safe in patients with increased cardiovascular risk. The potential effect on reducing cardiovascular events requires longer prospective studies and comparison with other drugs such as methotrexate, other TNF inhibitors, IL-17 and / or IL-23 antagonists. This should be added to the limitations of the study.

Response 5: We acknowledge the reviewer’s point. Following the reviewer’s request, we have now added in the Conclusion section the following phase:

Page 12, lines 453-4: “These findings suggest that apremilast is safe in patients with increased cardiovascular risk.”

In addition, we have now added in the Limitations the following phrase:

Page 9, lines 314-7: “Moreover, the potential effect of apremilast on reducing cardiovascular events requires longer prospective studies and comparison with other drugs such as methotrexate, other TNF inhibitors, IL-17 and/or IL-23 antagonists.”

Reviewer 2 Report

The authors show data supporting that apremilast, a PDE 4 inhibitor, protects the endothelial glycocalyx.  Apremilast is a drug to treat psoriasis.  An intact  endothelial glycocalyx is crucial in the determination of vascular permeability, attenuates blood cell and endothelium interaction and mediates shear stress sensing.

150 patients were enrolled and three psoriasis treatments (apremilast, etanercept, cyclosporine) were compared in this manuscript. A baseline was established and a fllow-up after 4 months of treatment.  Four parameters were measured perfused boundary region, sublingual microvessel diameter, functional microvascular density and global longitudinal strain.   Only apremilast had a positive effect on the perfused boundary region and an increase in functional microvascular density as well as pulse wave velocity.

The methods used are appropriate.  The data is strong.  The conclusion is apropriate.

Comments:  No comments

Author Response

Response to Reviewer 2 Comments

Comments:  No comments

Response: We would like to thank this Reviewer for approving the scientific comment of our study.

Reviewer 3 Report

This is a very valuable and interesting study that clearly demonstrates the beneficial effect of the phosphodiesterase 4 inhibitor (apremilast) on the cardiovascular system. The article is suitable for publication.

I have only one comment: 

- Please add a figure/table summarizing the possible effects of apremilast and the molecular mechanisms.

Author Response

Response to Reviewer 3 Comments

Point 1: Please add a figure/table summarizing the possible effects of apremilast and the molecular mechanisms.

Response 1: We would like to thank this Reviewer for approving the scientific comment of our study.

Following the reviewer’s suggestion, we have now added Figure 2 (revised manuscript, page 8) summarizing the molecular mechanisms of action of apremilast and the possible effects on cardiovascular system.